# Detection of six soil-transmitted helminths in human stool by qPCR- a systematic workflow

Kristy I. Azzopardi[1]*, Myra Hardy[1,2], Ciara Baker[1], Rhian Bonnici[1], Stacey Llewellyn[3], James S. McCarthy[3¤], Rebecca J. Traub[4], Andrew C. Steer[1,2]

1 Tropical Diseases Research Group, Murdoch Children's Research Institute, Melbourne, Victoria, Australia, 2 Department of Paediatrics, University of Melbourne, Melbourne, Victoria, Australia, 3 Clinical Tropical Medicine Laboratory, QIMR Berghofer Medical Research Institute, Brisbane, Queensland, Australia, 4 Faculty of Veterinary and Agricultural Sciences, University of Melbourne, Melbourne, Victoria, Australia

¤ Current address: The Peter Doherty Institute for Infection and Immunity, University of Melbourne, Melbourne, Victoria, Australia

* kristy.azzopardi@mcri.edu.au

**Data Availability Statement:** All relevant data are within the manuscript and its Supporting information files.

## Abstract

Soil-transmitted helminths (STH) infect up to one-quarter of the global population, with a significant associated disease burden. The main human STH are: *Ancylostoma* spp. and *Necator americanus* (hookworms); *Ascaris lumbricoides*, *Trichuris trichiura*, and *Strongyloides stercoralis*. The aim of this study was to establish a scalable system for stool STH multiplex quantitative real-time polymerase chain reactions (qPCR). Stool samples collected in Fiji and preserved in potassium dichromate were transferred to Melbourne at ambient temperature. Samples were washed to remove potassium dichromate and DNA was extracted with the Mini-Beadbeater-24 and a column-based kit. A SYBR green qPCR to detect the vertebrate mitochondrial gene was used as a DNA extraction control. Samples were tested using a probe-based multiplex qPCR targeting *A. lumbricoides*, *T. trichiura* and *S. stercoralis*, and in a second multiplex reaction to detect hookworms to the species level (*A. duodenale*, *A. ceylanicum*, *N. americanus*). An internal amplification control in both multiplex assays was included to prevent false-negative results due to PCR inhibitors. Samples were homogenised for a single cycle of 40 seconds to release STH DNA and washed stool was stored for up to 15 weeks at -30˚C without compromising DNA. Our multiplex qPCR detected multiple species of STH without reduced sensitivity compared to singleplex. qPCR data from 40 stools was validated against STH-positive stools determined by microscopy. We have developed and validated an efficient and staged system for detecting six clinically important STH affecting humans that could be easily implemented without advanced automation in any qPCR-capable laboratory.

## Introduction

Soil-transmitted helminths (STH) are intestinal parasites that infect 24% of the world's population causing a considerable global health burden [1]. Transmission of all STH involves contamination of soil by faeces containing parasitic eggs and/or larvae, and is the reason why

**Funding:** The study was funded by the Bill and Melinda Gates Foundation. The Murdoch Children's Research Institute is a non-for profit charity. No pharmaceutical grants were received in the development of this study.

**Competing interests:** The authors have declared that no competing interests exist.

infections are most common in communities with poor hygiene and sanitation [2]. The main STH species affecting humans are the hookworms *Ancylostoma duodenale* and *Necator americanus*; the roundworm *Ascaris lumbricoides*; the whipworm *Trichuris trichiura*; and the threadworm *Strongyloides stercoralis* [3]. There has been recent recognition that a third zoonotic hookworm, *Ancylostoma ceylanicum*, is also responsible for human infections in Asia [4–6].

To control STH infections, the World Health Organization [7] recommends bi-annual anthelminthic mass drug administration (MDA) when the prevalence of any STH exceeds 50%, and annual therapy when the prevalence is between 20% and 50% [8]. Effective control begins with accurate diagnosis, treatment, and follow-up using standardised procedures that can be easily adopted across a variety of laboratories. The most commonly used method for detection of *Ascaris, Trichuris* and the hookworms is the Kato-Katz (KK) thick smear in which helminth eggs are identified and enumerated by microscopy [9]. The KK method is recommended by WHO and allows for quantification whereby parasitic load is expressed as eggs per gram of stool (EPG) [8]. The sensitivity of the KK method varies from 65.2–96.9%, while specificity is high (93.8–99.4%), [10–15]. Other techniques include concentration, for example, by the McMaster method or by using automated devices such as FLOTAC and mini-FLOTAC. The sensitivity and specificity of these techniques range from 54.0–99.4% and 91.3–99.6%, respectively [3, 16–19]. However, all of these four mentioned methods are unable to reliably diagnose *S. stercoralis*, as only larvae are found in the stool [13]. The Baermann's procedure has been proposed as the gold standard for diagnosis of *Stronglyloides* and relies on active migration of *S. stercoralis* larvae from stool into a water phase via a funnel [3]. Sensitivity is imperfect (28.3–72.0%) due to the heterogeneity in the distribution of larvae output in stool, whilst specificity of this assay is reported to be between 75.2–100% [13, 20, 21]. Although the material costs of these tests are modest, there are multiple drawbacks, including: the requirement of a trained parasitologist for microscopic examination, low sensitivity in endemic populations with low intensity infection, the need to assess the stool immediately after collection, and the inability to diagnose down to the species-level for the hookworms [22].

Molecular techniques such as quantitative real-time polymerase chain-reaction (qPCR) can overcome many of the disadvantages associated with classical microscopy. The generation of sophisticated chemistries incorporated into convenient master mixes has made qPCR one of the most powerful technologies in molecular microbiology, and furthermore, different coloured fluorophores conjugated to target probes allow for the detection of multiple targets within one reaction (multiplex), making the assay cost-effective (saving of reagents and run-time), and highly specific. Studies have shown that qPCR is more sensitive than microscopy and conventional PCR for the detection of various intestinal helminths [4, 23–26]. By designing primers against the conserved and abundant 18S rRNA gene (as opposed to single-copy DNA targets), a high level of specificity and sensitivity can be achieved for *S. stercoralis* and *T. trichiura* [27, 28]. Specificity of the *S. stercoralis* qPCR developed by Verweij et al was reported as 100% and could detect infection in 12 of 158 human stool samples from Ghana that were previously negative by the Baermann method [28]. A study investigating STH infections in Philippine school children showed that *Ascaris lumbricoides* detection by qPCR outperformed diagnosis by KK thick smears performed in triplicate (61% versus 21%, respectively) [29], and similarly, detection of *T. trichiura* by qPCR outperformed KK in human stool from Flores Island, Indonesia (51.7% versus 45%, respectively) [30]. Hii *et al* developed a multiplex qPCR to simultaneously detect and discriminate between all three hookworm species found in human stool (*N. americanus*, *A. ceylanicum*, *A. duodenale*) making it an appealing tool for identification of mixed hookworm infections in STH-endemic communities [4]. Recently, a study of 648 stools from Myanmar reported that the prevalence of any

STH was 20.68% by KK and 45.06% by qPCR, with hookworm detection by qPCR exceeding KK by approximately 4-fold [26].

Unfortunately, there is no gold standard qPCR or harmonised protocols for detection of these major parasitic worms in stool [31, 32], and selecting a method is further confounded by reporting sections of the pipeline and not the entire set of processes. The workflow can be fragmented into three main stages: (i) Preservation and preparation of stool, (ii) DNA extraction and pre-testing, and finally (iii) Detection of the STH by qPCR. Firstly, stool is collected and preserved to avoid DNA degradation and deactivate nucleases found in faeces. Placing the sample immediately at -20˚C (or below) is ideal, however not often a possibility in remote settings. Ethanol has been validated for use as a preservative in the absence of cold chain [30, 33–35], but transport in large volumes by air becomes restricted and costly due to its classification as a dangerous good. Potassium dichromate is commonly used in STH field trials to provide long-term preservation of stool at ambient temperatures and does not have the same freight restrictions [5, 34, 36]. Once the potassium dichromate is washed off, the stool must undergo DNA extraction within 4 weeks otherwise the STH DNA may degrade due to removal of the preservative [36]. It remains unknown whether washed stool could be stored and processed for DNA extraction beyond this time. In the second stage, parasitic DNA is released from stool using chemical lysis (commercial DNA kit), in conjunction with mechanical disruption using a high-speed homogeniser or beadbeater. Several researchers have shown that bead-beating is essential when extracting helminth DNA from human stool but in the absence of a standardised protocol it remains unclear if some homogenizers are superior to others [30, 33, 37–39]. Furthermore, bead-beating times range from 30 seconds to 3 minutes [30, 36, 40] and whilst these may seem like minor details, the process of DNA extraction underpins the success of STH detection. The final stage of the workflow involves a series of qPCRs, often in multiplex format to achieve the biggest bang for buck.

The primary aim of this study was to detect six clinically important STH affecting humans using qPCR in stool preserved in potassium dichromate. We utilised a multiplex qPCR assay (designated STH1) with a panel of four fluorophores to capture DNA from *Ascaris lumbricoides*, *T. trichiura* and *S. stercoralis* and an internal amplification control (IAC), along with a second multiplex assay (designated STH2) for the hookworms *N. americanus*, *A. ceylanicum*, *A. duodenale* and an IAC, with minor modifications to the assay previously published [4]. Whilst evaluating this staged approach, we sought to fill these technical gaps with the overall aim of facilitating a more effective process. Specifically, we: (i) investigated whether washed stool could be stored below -30˚C to allow for DNA re-extraction; (ii) optimised the cycle time for homogenisation of stool during DNA extraction; and (iii) incorporated simple qualitative analysis of extracted DNA.

## Materials and methods

### Field samples

Forty stools were collected as part of the Fiji Integrated Therapy study, a three-arm, cluster randomised, open-label safety and efficacy trial that involved whole populations of two Fijian islands, Rotuma and Gau [41]. The study protocol was approved by the Royal Children's Hospital Melbourne Human Research Ethics Committee (reference 36205) and Fiji National Health Research and Ethics Review Committee (reference 2016·81·MC). Stool samples were processed within 12 hours of collection. Standard KK method was used to prepare slides in duplicate for microscopy identification and quantification. Slides were read independently by one of two trained laboratory technicians and considered positive if either technician identified eggs of *A. lumbricoides*, *T. trichiura* or hookworm. Counted eggs were multiplied by a

factor of 24 to account for a 41.7 mg stool template. Final EPG counts were the average of the two readers. Remaining sample was preserved with potassium dichromate and mixed 1:1 with 5% (weight/volume) potassium dichromate for a final concentration of 2.5% (w/v) potassium dichromate and stored at room temperature until testing. Samples were collected over a period of 16 months (July 2017 to November 2018) and then transported to The Murdoch Children's Research Institute (MCRI) laboratories where testing was performed between May 2018 to February 2019.

## Removal of potassium dichromate

Prior to DNA extraction, potassium dichromate was removed from stools by washing as previously described [36] but with modification of the centrifugation speed. Briefly, a wooden spatula was used to mix the preserved stool and up to 7.5 ml was poured into a 50 ml Falcon tube and centrifuged at 700 x*g* for 3 minutes, to avoid lysis of worm eggs. Potassium dichromate was poured into toxic waste containers, and the stool was washed with 50 ml of phosphate-buffered saline (PBS) and centrifuged again. Final washing was performed with 15 ml of PBS and the sample was then transferred into a 15 ml tube. Following centrifugation, the supernatant was removed leaving the washed stool pellet. Washed samples were stored at 4˚C until the DNA was extracted (within 4 weeks).

## DNA extraction and pre-testing checkpoint

DNA extraction was performed on 200 mg of washed stool using the Isolate II Fecal DNA Kit (Bioline, UK) according to the manufacturer's protocol, with bead beating for 40 seconds on the Mini-Beadbeater-24 at maximum speed (3800 rpm; Bio Spec Products, USA) and final elution in 100 μl of buffer. Following extraction, the remaining washed stool was stored at -30˚C. The FastPrep-24™ 5G (MP Biomedicals, USA) was compared to the Mini-Beadbeater-24 in the optimisation of DNA extraction tests. qPCR for the mitochondrial 16S rRNA gene was performed to test whether the extraction was successful. GoTaq® SYBR green qPCR Master Mix (Promega, USA) was used to amplify the vertebrate mitochondrial (mt) 16S ribosomal RNA gene (mt-F: CGACCTCGATGTTGGATCAG; mt-R: GAACTCAGATCACGTAGGACTTT) [42]. Reactions were performed with 500 nm of each primer, using 2 μl of DNA in a final volume of 10 μl. The Mx3005P (Agilent, Germany) thermal cycler was used for gene amplification: 1 cycle of 95˚C for 2 m; followed by 40 cycles of 95˚C for 3 s and 60˚C for 30 s; and final dissociation of 60–95˚C. Samples with quantification cycle ($C_q$s) >25 (determined using mean+3SD from DNA extracts of 40 stools) were re-extracted using frozen-washed stool, and tested again (S1 Fig). DNA from healthy stools (triplicate) was extracted and served as a negative control for each optimisation experiment. Pure DNA samples were stored at -80˚C.

## Storage of washed stools at -30˚C

DNA was extracted from 12 stools following washing and storage at 4˚C for up to 4 weeks. The same washed stools were then stored at -30˚C for various lengths of time: 1 week (n = 6), 5 weeks (n = 4) and 15 weeks (n = 2); and DNA was extracted again. SYBR green-based qPCR was performed on paired DNA samples in duplicate, to detect the mt 16S rRNA gene and resulting $C_q$s were compared.

## Comparison of homogenisers

*S. stercoralis*-positive stool (confirmed by microscopy but not quantified) was spiked with *A. lumbricoides* eggs, and the sample was divided into eight aliquots of 200 mg each containing

approximately 1000 *A. lumbricoides* eggs. DNA was extracted using the Isolate II Fecal DNA Kit according to the manufacturer's protocol, with bead-beating performed on both disruptors using four different cycles: 40 seconds x1 cycle; 40 seconds x2 cycles; 40 seconds x3 cycles and 30 seconds x4 cycles. qPCR for the parasites was performed in triplicate (S1 Table). Hookworm and *Trichiura* eggs were not available for comparison.

## Multiplex STH qPCR reactions

Reactions were performed in duplicate using the GoTaq® qPCR Probe Master Mix (Promega, USA) with 5 µl of stool DNA in a final reaction volume of 25 µl. Final primer and probe concentrations for STH1 and STH2 were modified slightly from original sources by reducing primer and probe concentrations (Table 1). Cycle conditions were: 1 cycle of 95˚C for 5 minutes; followed by 40 cycles of 95˚C for 10 seconds and 60˚C for 60 seconds. To assess reaction dynamics as single versus multiplex qPCRs, seven-point standard curves were generated for each STH using 10-fold diluted ($2.5 \times 10^2$ to $2.5 \times 10^{-4}$ pg) synthetic genes (gBlocks®; Integrated DNA Technologies® or IDT®, USA) designed to mimic the target DNA sequences (S2 Table). These gBlocks® were also used as positive controls for validation of field samples.

## Detection of *Trichuris* in spiked-stools

Healthy stools were spiked, in quadruplicates, with known amounts of *Trichuris suis* eggs ranging from 0 to 640 EPG of stool (*T. suis* 18S sequence homologous to *T. trichiura*) and divided into two cohorts. This first group was extracted as per our protocol, and the second

**Table 1. The primers and probes used in the two multiplex reactions.**

| qPCR | Target | Primer/Probe | Sequence (5'-3') | Size (bp) | Gene target | Final conc (nM) | Source |
|---|---|---|---|---|---|---|---|
| STH1 | *Strongyloides stercoralis* | Strongy-F | GAATTCCAAGTAAACGTAAGTCATTAGC | 101 | 18S | 100 | *Modified from [28] |
| | | Strongy-R | TGCCTCTGGATATTGCTCAGTTC | | | 100 | |
| | | Strongy-P | FAM-ACACACCG/ZEN/CCGTCGCTGC-IBFQ | | | 50 | |
| STH1 | *Trichuris trichiura* | Tri-F | TTGAAACGACTTGCTCATCAACTT | 76 | 18S | 100 | *Modified from [27] |
| | | Tri-R | CTGATTCTCCGTTAACCGTTGTC | | | 100 | |
| | | Tri-P | CY5-CGATGGTAC/TAO/GCTACGTGCTTACCATGG-IBRQ | | | 50 | |
| STH1 | *Ascaris lumbricoides* | Asc-F | GTAATAGCAGTCGGCGGTTTCTT | 87 | Internal transcribed spacer 1 | 100 | *Modified from [43] |
| | | Asc-R | GCCCAACATGCCACCTATTC | | | 100 | |
| | | Asc-P | HEX-TTGGCGGAC/ZEN/AATTGCATGCGAT-IBFQ | | | 50 | |
| STH2 | *Ancylostoma* spp | Anc-F | CGGGAAGGTTGGGAGTATC | 104 | Internal transcribed spacer 1 | 100 | *Modified from [4] |
| | | Anc-R | CGAACTTCGCACAGCAATC | | | 100 | |
| | *A. duodenale* | Aduo-P | HEX-TCGTTAC+T+GGGTGACGG-IBFQ | | | 50 | |
| | *A. ceylanicum* | Acey-P | FAM-CCGTTC+CTGGGTGGC-IBFQ | | | 50 | |
| STH2 | *Necator americanus* | Nec-F | CTGTTTGTCGAACGGTACTTGC | 101 | Internal transcribed spacer 2 | 100 | [44] |
| | | Nec-R | ATAACAGCGTGCACATGTTGC | | | 100 | |
| | | Nec-P | CY5-CTG+TA+CTA+CG+CAT+TGTATAC-IBRQ | | | 50 | |
| STH1 & STH2 | Equine herpesvirus type 4 | EHV-F | GATGACACTAGCGACTTCGA | 81 | Glycoprotein B | 40 | [45] |
| | | EHV-R | CAGGGCAGAAACCATAGACA | | | 40 | |
| | | EHV-P | ROX-TTTCGCGTGCCTCCTCCAG-IBRQ | | | 100 | |

+: Locked nucleic acid (LNA); IBFQ and IBRQ: Iowa Black® Dark Quenchers (Integrated DNA Technologies®).

*Modification of primer concentrations.

underwent DNA isolation using the QIAamp Powerfecal DNA Kit (Qiagen) incorporating 0.7 mm garnet beads.

## Data analysis

Differences in $C_q$s obtained for storage of washed stools were assessed using the Wilcoxon matched-pairs signed rank test, and for analysis of homogenisers and cycle times we applied the paired t test. Acceptable qPCR standards produced $R^2$ values $\geq 0.95$ and PCR efficiencies between 90–110%. Equine herpesvirus type 4 (EHV-4) DNA was used in each reaction as an IAC. Samples with EHV-4 $C_q$ values $\geq 2$ cycles compared to the EHV $C_q$ of the negative control, were deemed a failure and the qPCR was repeated. Wells were considered negative if the $C_q$ values were $\geq 37$ as water-only controls intermittently produced false positives above this $C_q$. A sample was considered positive by qPCR if either STH1 or STH2 registered a pathogen with $C_q < 37$. Due to the inherent limitations of specification with KK, when we compared KK and qPCR results, we considered any positive qPCR result for *Ancylostoma duodenale*, *Ancylostoma ceylanicum* or *Necator americanus* to be hookworm positive, and *S. stercoralis* results were excluded from the comparison. We used direct method comparisons and a pseudo-gold standard of combined results from both methods to report the diagnostic accuracy [13], and Cohen's kappa statistics to assess level of agreement.

## Results

### Summary of laboratory workflow

Our laboratory procedures (Fig 1) were developed following a series of experiments aimed at optimising the three main steps of the process: (i) Preservation and preparation of stool, (ii) DNA extraction and pre-testing, and (iii) Detection of the unique combination of six important STH using two multiplex qPCRs.

### Storage of washed stools at -30˚C

The data show that no significant change in $C_q$s (*p*0.0670) was observed for washed stools stored at -30˚C compared to stools stored at 4˚C for up to 4 weeks (mean $C_q$ = 20.37 vs 20.62, respectively) indicating that the amount of DNA extracted was not reduced (Fig 2).

### Comparison of homogenisers

We compared two high-energy cell disrupters for homogenisation of stool positive for both *S. stercoralis* (fragile larvae, easy to lyse) and *A. lumbricoides* (hard egg shell), using the Mini-Bead-beater-24 (BB) and the FastPrep-24™ 5G (FP). We also tested whether increasing homogenisation cycles could improve recovery of parasitic DNA (determined by STH1 qPCR $C_q$). No difference in machine performance was observed for either parasite (*S. stercoralis* 95% confidence interval (CI) = -0.7286 to 1.427, *p* = 0.3784; *A. lumbricoides* 95% CI = -0.7012 to 1.496, *p* = 0.3330) (Fig 3 and S1 Table). Increasing homogenisation cycle times on the FastPrep-24™ 5G produced lower $C_q$s for *S. stercoralis* ($C_q$ range 25.87–27.72), but the same trend was not observed when using the Mini-Beadbeater-24 ($C_q$ range 26.62–27.74). Improved detection of *A. lumbricoides* was achieved with the Mini-Beadbeater-24 using 1 cycle of homogenisation for 40 s (mean $C_q$ 32.29) and future experiments implemented this cycle on the Mini-Beadbeater-24.

### Multiplex STH qPCR reactions

We combined published primers and probes to design a multiplex assay (STH1), for simultaneous detection of *S. stercoralis*, *T. trichuria*, and *A. lumbricoides* [27, 28, 43], whilst the second

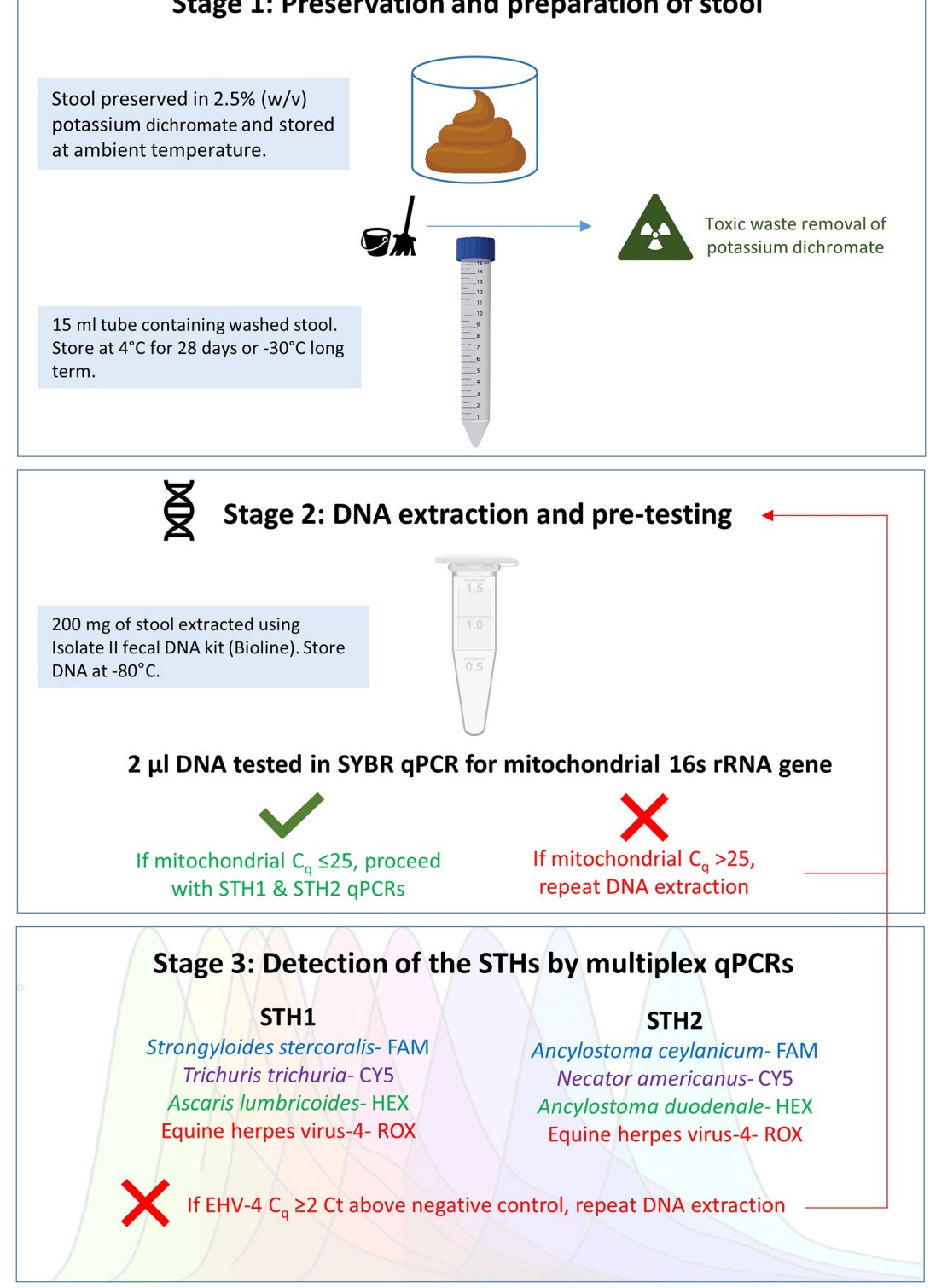

**Fig 1. Workflow of the laboratory procedures.** The three-staged approach for detection of soil-transmitted helminths in potassium dichromate preserved stool. Images used under license from Shutterstock.com.

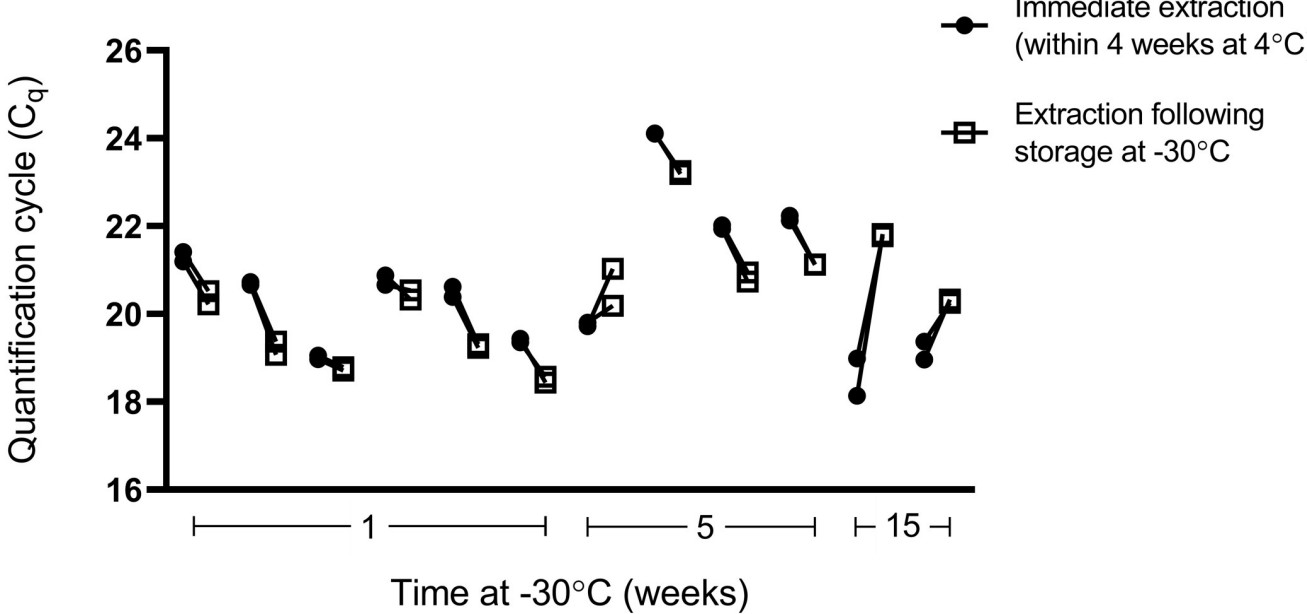

**Fig 2. Effects of storing washed stools at -30˚C on DNA.** SYBR green qPCR for the vertebrate mitochondrial gene was used to determine DNA integrity following "immediate" extraction from stools (stored 4˚C for $\leq$4 weeks, ●) versus stools washed and stored at -30˚C (□) for different lengths of time: 1 w (n = 6), 5 w (n = 4) and 15 w (n = 2). Mean $C_q$s and standard deviations are shown for duplicate reactions.

hookworm multiplex assay (STH2) was performed as published with minor modifications [4]. Both qPCRs were performed in singleplex and multiplex formats using synthetic gBlock® DNA to ensure that sensitivity of gene amplification was not lost due to competition in the multiplex PCR targeting the three parasites (Fig 4). The quantification cycles, efficiencies and $R^2$ values from both assays were similar and the same analytical sensitivity (or limit of detection) of 2.4 x $10^{-4}$ pg of gBlock® per reaction was achieved for both formats (Table 2).

### Assay validation using field samples

A subset of KK microscopy positive and negative stools (n = 40) from Fiji were randomly selected for testing using our protocol (Fig 1). Vertebrate mt $C_q$s for the samples varied from 16.93–23.97 with a mean of 19.71 (S1 Fig). Multiplex qPCR detected *A. lumbricoides*, *S. stercoralis*, *A. ceylanicum* and *N. americanus*, but not *A. duodenale* and *T. trichiura* (Table 3 & S3 Table). Absence of any qPCR positive *Trichuris* reduced the overall sensitivity of qPCR to 50% compared to KK, and 64.3% compared to the combined pseudo-gold standard. The sensitivity for *A. lumbricoides* diagnosis was 75% and 77.8% respectively and for hookworm it was 66.7% and 82.4%. The agreement between the two methods was again low overall due to the zero agreement with *T. trichiura* samples but was fair for hookworm and substantial for *A. lumbricoides*. Two of four samples with polyparasitism by microscopy were confirmed by qPCR. An additional two samples that were microscopy negative had polyparasitism detected by qPCR. Importantly within each multiplex assay we were able to detect co-infection; one sample *S. stercoralis* and *A. lumbricoides* positive and another *A. ceylanicum* and *N. americanus* positive.

### Detection of *Trichuris* in spiked-stool

To further explore why the STH1 qPCR was unable to detect *T. trichiura* in any of the six stools diagnosed positive by KK (EPG range 48–384, mean 170), we re-explored our DNA extraction method specifically targeting the ability to lyse the robust *T. trichiura* eggs. This has been

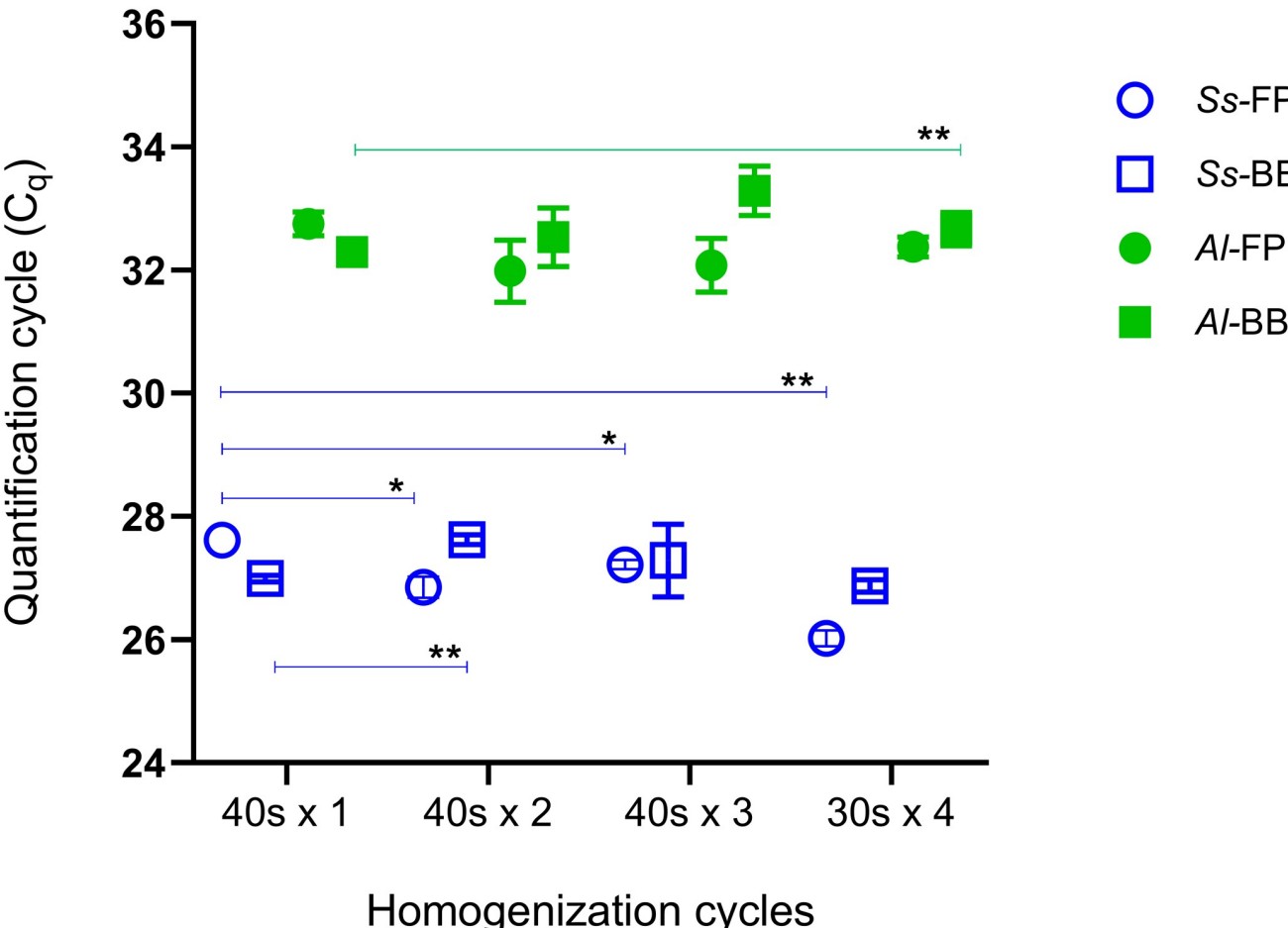

**Fig 3. Comparison of two homogenisers for DNA extraction.** The FastPrep-24™ 5G (FP) and Mini-Beadbeater-24 (BB) were used in conjunction with the Isolate II Fecal Kit to detect *S. stercoralis* (*Ss*) larvae and *A. lumbricoides* (*Al*) extracted from human stool by qPCR. Stool was equally divided into 8 aliquots and extraction was performed on either the FP or BB. Four different cycling times were tested using each homogeniser and the STH1 qPCR was performed in triplicate to ascertain mean $C_q$s and standard deviations. *p<0.05; **p<0.005 using paired t tests.

identified as a problem previously [30, 46, 47] with garnet beads for bead-beating identified in one study as the most effective for isolation of *T. trichiura* DNA [30]. The STH1 qPCR detected *T. suis* in stools containing 160, 320 and 640 EPG of stool using both kits, with similar $C_q$s obtained at 320 and 640 EPG (range 30.58–31.98) (Fig 5). The QIAamp Powerfecal kit produced lower $C_q$s (mean 33.37) at 160 EPG compared to the Bioline kit (mean $C_q$ 36.41), however the limit of *T. suis* detection using both kits was 160 EPG. Overall, no significant difference in $C_q$s was observed between the kits (*p* = 0.2446). Notwithstanding this observed increased sensitivity provided by the QIAamp Powerfecal kit, these data confirmed that the Bioline kit was able to detect *T. suis* down to 160 EPG of stool. A previous study in Liberia reported that the barrel-shaped eggs of *Capillaria* species could be mistaken for *T. trichiura* using KK [48]. We were able to exclude this as a cause of our low sensitivity by performing an 18S PCR which did not identify any *Capillaria* species.

## Discussion

We developed a staged workflow to detect the presence of six important STHs that infect humans using two multiplex qPCRs. Stools were subjected to homogenisation using the

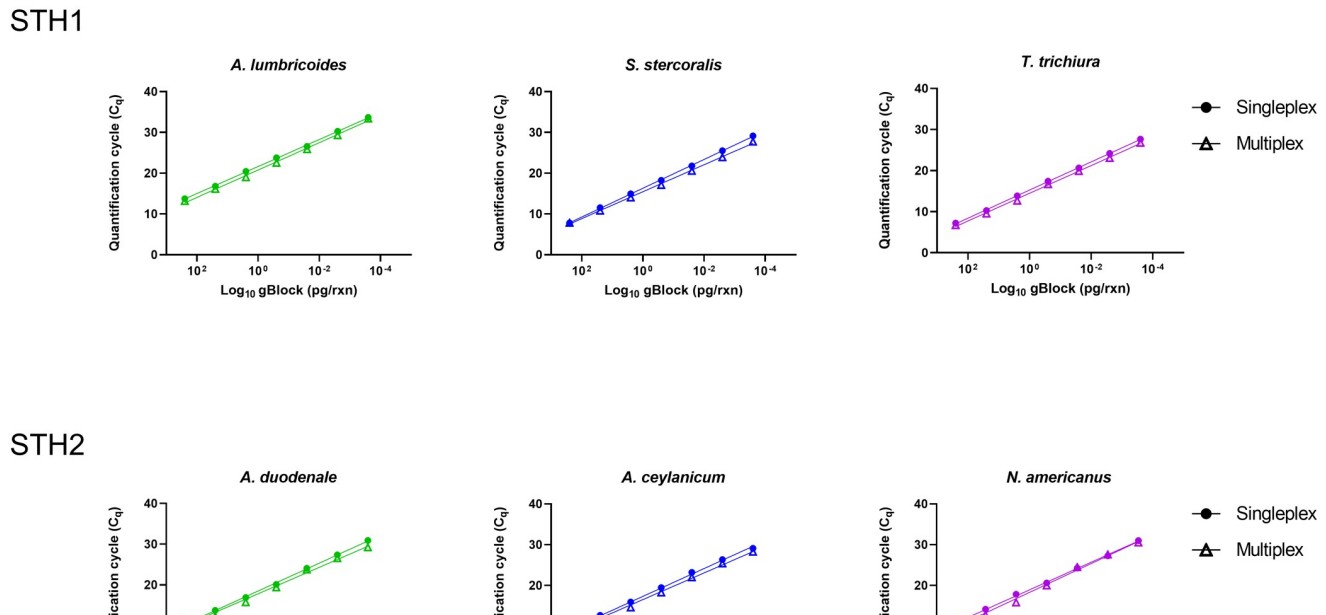

**Fig 4. The STH1 and STH2 qPCRs performed as both singleplex and multiplex reactions.** Synthetic gBlocks® (IDT) mimicking sequences from each of the six parasites were diluted 10-fold to generate 7-point standard curves. Mean $C_q$ values and standard deviations from duplicate wells are shown.

Mini-Beadbeater-24 and purified using a column-based kit from Bioline. We incorporated a quality checkpoint to validate extracted DNA using SYBR qPCR targeting a vertebrate mitochondrial gene. If $C_q$s were ≤25, DNA underwent final testing in the STH1 and STH2 multiplexes, and if $C_q$s were >25, DNA isolation was repeated. We successfully configured STH1 to

**Table 2. Analytical performance of the two multiplex assays.**

|  | STH1 | | STH2 | |
|---|---|---|---|---|
|  | **Singleplex** | **Multiplex** | **Singleplex** | **Multiplex** |
|  | *A. lumbricoides* | | *A. duodenale* | |
| **$R^2$** | 0.999 | 0.995 | 0.998 | 0.9855 |
| **Efficiency (%)** | 100.3 | 98.7 | 96.3 | 103.4 |
| **Formula** | Y = -3.315X + 21.63 | Y = -3.355X + 20.77 | Y = -3.413X + 18.48 | Y = -3.244X + 17.79 |
|  | *S. stercoralis* | | *A. ceylanicum* | |
| **$R^2$** | 0.999 | 0.998 | 0.997 | 0.990 |
| **Efficiency (%)** | 92.1 | 100.8 | 96.47 | 102.23 |
| **Formula** | Y = -3.528X + 16.30 | Y = -3.303X + 15.46 | Y = -3.409X + 17.32 | Y = -3.270X + 16.53 |
|  | *T. trichiura* | | *N. americanus* | |
| **$R^2$** | 0.998 | 0.998 | 0.993 | 0.992 |
| **Efficiency (%)** | 95.9 | 98 | 95.8 | 95.3 |
| **Formula** | Y = -3.424X + 15.26 | Y = -3.371X +14.47 | Y = -3.426X + 18.79 | Y = -3.441X + 18.16 |

PCR efficiencies and linear dynamic range for STH1 and STH2 in both formats (Fig 4) are described above. Reactions were performed in a total volume of 25 μl containing 5 μl of synthetic gBlock® DNA (IDT). The MX3005P real-time PCR system was used in this study.

**Table 3. Validation of the STH1 and STH2 qPCRs.**

| Result | Positive | | | Negative | Sensitivity qPCR compared to KK % | Kappa agreement | Sensitivity qPCR compared to combined[a] % |
|---|---|---|---|---|---|---|---|
| | qPCR only | KK only | Both | Both | | | |
| **Any parasite**[b] | 8 | 8 | 10 | 14 | 55.6 | 0.19 | 69.2 |
| **STH1** | | | | | | | |
| *A. lumbricoides* | 1 | 2 | 6 | 31 | 75.0 | 0.75 | 77.8 |
| *S. stercoralis* | 1 | Nd | Nd | Nd | Nd | Nd | Nd |
| *T. trichiura* | 0 | 6 | 0 | 34 | 0 | 0 | 0 |
| **STH2** | | | | | | | |
| Hookworms[c] | 8 | 3 | 6 | 23 | 66.7 | 0.34 | 82.4 |
| *A. duodenale* | 0 | | 0 | | | | |
| *A. ceylanicum* | 5 | | 2 | | | | |
| *N. americanus* | 3 | | 5 | | | | |

KK: Kato-Katz microscopy; Nd: Not done; Cohen's Kappa statistic agreement: 0 = agreement equivalent to chance, 0.1–0.2 = slight agreement, 0.21–0.40 = fair agreement, 0.41–0.60 = moderate agreement, 0.61–0.80 = substantial agreement, 0.81–0.99 = near perfect agreement.

[a] Combined is the pseudo-gold standard of results from both methods.

[b] Including mixed infections.

[c] Any hookworm (KK microscopy cannot distinguish between species).

simultaneously detect three human helminths- *A. lumbricoides*, *S. stercoralis*, *T. trichiura*, without loss of sensitivity or specificity when compared to singleplex reactions. Both qPCRs incorporated an internal amplification control using EHV-4 to ensure that negative results were real and not due to reaction inhibition. qPCR data from 40 stools were validated against STH-positive stools determined by Kato-Katz.

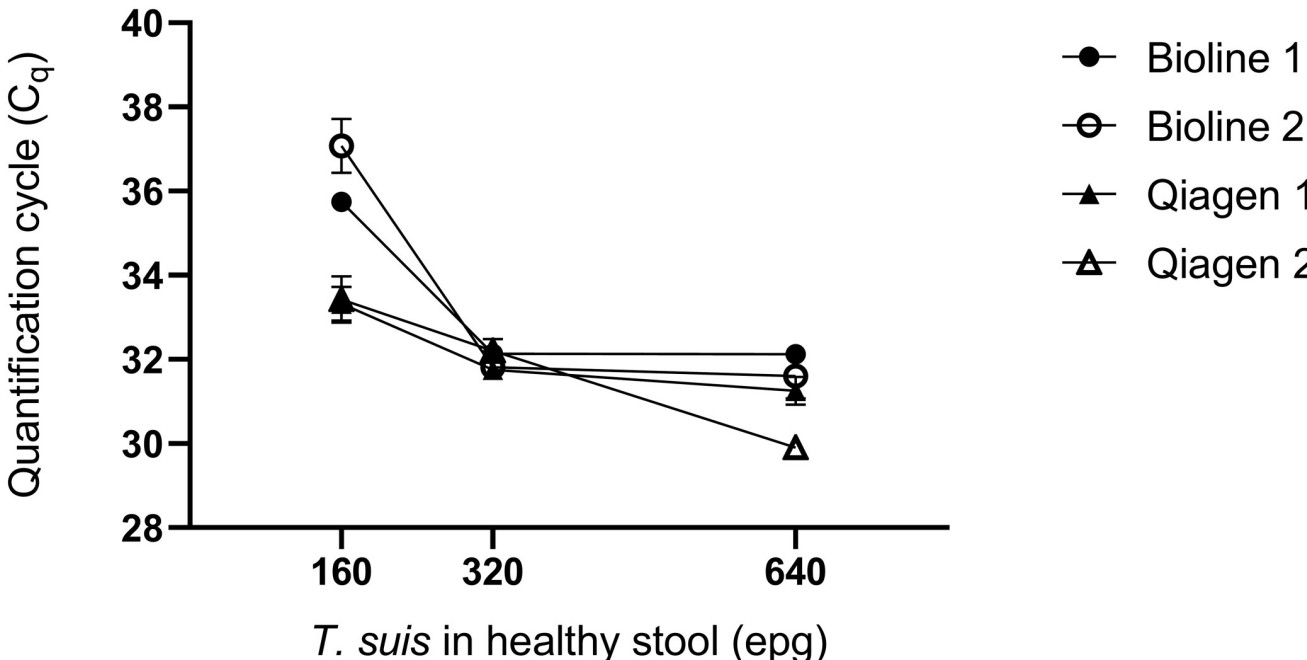

**Fig 5. *Trichuris suis* spiked stool experiment.** Healthy stool was spiked with *Trichuris suis* eggs ranging from 0 to 640 EPG of stool and divided equally into four groups (two replicates per kit). DNA was extracted using either the Bioline fecal kit (per our protocol) or the Qiagen Powerfecal Kit incorporating 0.7 mm garnet beads. Each symbol is the mean and standard deviation from duplicate qPCR reactions.

Our protocols require no microscopic examination and could be easily implemented without advanced automation in any qPCR-capable laboratory. Potassium dichromate permits preservation at room temperature in remote locations, until stools arrive at the laboratory for molecular testing. We demonstrated that washed stools could be stored at -30˚C before DNA isolation for up to 15 weeks without compromising yield. This finding enables an improved workflow, if extraction failed and requires repeating, it would be easier to return to the washed sample instead of repeating the washing process.

For our samples, DNA was prepared using the Isolate II fecal DNA kit, which has been used in a similar setting to detect parasitic DNA in stools [4]. This column-based method offers a cost-effective and simple protocol for manual handling which would suit small laboratories that do not have access to robotic technology. We compared two high-energy cell disruptors available to us, with different cycle conditions for the extraction of *S. stercoralis* and *A. lumbricoides*. These parasites represent different cell morphologies—*S. stercoralis* has fragile larvae that are easy to lyse, while *A. lumbricoides* is known to possess a rigid egg shell. Using human stool containing *S. stercoralis* larvae and *A. lumbricoides* eggs, we were able to demonstrate that one cycle of homogenisation for 40 seconds on the Mini-Beadbeater-24 was just as effective as the FastPrep-24™ 5G with longer cycle times [30, 49]. Our method has minimised the time allocated to this task without compromising the DNA extracted. However, as technology and kits advance, extraction methods may continue to improve as evident by a recent study that found improved sensitivity and DNA yield of *T. trichiura* and the *N. americanus* when the Qiagen Blood & Tissue kit was used with bead-beating compared to the Qiagen Stool kit [33].

To validate the quality of extracted DNA before continuing onto the multiplex qPCRs, we implemented the first quality control checkpoint in stage 2 of our workflow—amplification of a vertebrate 16S rRNA mitochondrial gene. To rule out the presence of inhibitors within stool and the possibility of potassium dichromate-carryover, we used the mt qPCR as a proxy for assessing the extracted DNA prior to multiplexing. Use of SYBR-based chemistry in a small reaction volume of 10 μl minimised costs and expended only 4 ul of stool DNA (2 ul in duplicate).

Our STH1 qPCR detected the presence of *A. lumbricoides* and *S. stercoralis* in a single stool sample, whilst the STH2 qPCR confirmed a dual-hookworm infection with *A. ceylanicum* and *N. americanus*. This increased detail in species identification will assist in understanding site epidemiology and opportunities for intervention, both prevention and treatment.

Another advantage of this workflow is that multiplexing saves on reagents, consumables and the DNA sample itself. Processing time is reduced because individual PCRs do not need to be performed. The EHV-4 internal amplification control in stage 3 provided additional quality assurance of sample preparation in real-time. Each qPCR reaction was spiked with the viral gBlock® (synthetic DNA) and detected on the fourth ROX channel. In our 40 samples, the EHV-4 $C_q$ in each well did not deviate by more than 2 $C_q$s above the negative control wells, giving us confidence that our systematic approach yielded DNA that could be reliably amplified using robust qPCR reagents manufactured to resist PCR inhibitors.

In our small pilot study, we were able to achieve high sensitivity for *A. lumbricoides* and hookworm diagnoses compared to the pseudo-gold standard of combined positive results. The overall level of agreement was heavily impacted by the absence of *T. trichiura* detection using qPCR. Possible explanations for the poor sensitivity for detection of T. *trichiura* is the low infection intensity in our samples, since most samples were at or below the 160 EPG threshold of detection established in our spiked *T. suis* experiment. Validation of the STH1 multiplex using gBlocks® and *T. suis* provide confidence that the STH1 multiplex is capable of detecting *T. trichiura*. We report up to 3 $C_q$s difference between DNA extraction kits (Fig 5), therefore it

is possible that an alternative DNA extraction kit, not tested in this study, could improve detection of low intensity *T. trichiura* infections. The absence of detection of *A. duodenale* is likely related to the local epidemiology of hookworms. Parasites detected by KK but not qPCR might be attributed to the over 4-fold increase in stool volume used for KK (41.7 mg) compared to qPCR (10 mg). We tried to limit this effect by maximising the volume of DNA tested (5 ul per reaction, in duplicate) in our multiplex qPCRs to achieve the highest sensitivity without compromising the reaction by introducing inhibitors. Work is currently underway to test a larger sample size. This will allow us to further explore the practicality of this molecular workflow, and to more fully investigate the sensitivity and specificity of our approach compared to microscopy.

A possible limitation in this study is the use of potassium dichromate as a preservative. It is important to note that in the Myanmar study, which reported higher detection of *T. trichiura* by qPCR than KK, stool was placed directly at -20˚C and not preserved prior to molecular analysis [26]. It is possible that the potassium dichromate in our study could have prevented detection of *T. trichiura* by qPCR, as the effects of this preservative was not accounted for in our *T. suis* experiment. When compared to potassium dichromate, RNA *later* and 96% ethanol have been reported to increase DNA yield for *A. lumbricoides*, *T. trichiura* and *N. americanus* [33]. In future, we could trial different DNA and RNA preservatives that are commercially available. These reagents are available in kits, are nontoxic, and can be easily deployed to a remote setting with tropical climates, but they are likely to be more expensive. The requirement for multiple rinses to remove the potassium dichromate may have also resulted in loss of free helminth DNA from degraded eggs, however most would be kept intact within eggs or larvae until DNA extraction was performed. Another limitation is the use of the pseudo-gold standard for comparison of qPCR and KK data, as the pseudo-gold standard is weighted in favour of the qPCR (because of inclusion of its positive results); yet this type of analyses has been used in the absence of a true gold standard [12, 13, 50, 51]. Another challenge we faced, which is shared by others in the field, is the difficulty in obtaining worm eggs to use as controls [31, 32]. In order to implement a universal qPCR to diagnose these STH, there is a need for a global bank of eggs, larvae and DNA to be made available as there is for other parasites.

## Conclusions

We have developed a scalable system to wash stool preserved in potassium dichromate and extract DNA using a simple column-based kit with a single bead-beating cycle. Quality-control checkpoints were implemented to avoid false-negative results. This is the only study to present a qPCR workflow to detect this unique combination of six important STHs affecting 1.4 billion humans worldwide [32]. We utilised two multiplex qPCR reactions to detect *Ancylostoma duodenale* and *ceylanicum*, *Necator americanus*, *Ascaris lumbricoides*, *Trichuris trichiura* and *Strongyloides stercoralis*. In the absence of a 'gold standard' for diagnosis of STH infections by qPCR, this work could be very useful towards development of a standard diagnostic test. The methods can be applied for samples collected from remote settings and processed in low resourced molecular laboratories where infections with these parasites thrive. This approach has the potential to improve diagnosis of STH for surveillance, mapping and field studies in areas where STH are endemic and in need of public health control.

## Supporting information

**S1 Fig. qPCR of 40 DNA extracts for the vertebrate mitochondrial 16S rRNA gene.** Stool DNA with $C_q$s over 25 (mean+3SD, dotted line) underwent repeat DNA extraction. Mean

with standard deviation derived using duplicate qPCRs.
(TIF)

**S1 Table. Comparison of two high-energy cell disruptors for DNA extraction.** The FastPrep-24™ 5G (FP) and Mini-Beadbeater-24 (BB) were used in conjunction with the Isolate II Fecal Kit to detect *S. stercoralis* (*Ss*) larvae and *A. lumbricoides* (*Al*) extracted from human stool by qPCR. Mean $C_q$ values and standard deviations from triplicate wells are shown.
(PDF)

**S2 Table. The gBlock® gene sequences used in this study.**
(PDF)

**S3 Table. Performance of the vertebrate qPCR, and the STH1 and STH2 multiplexes.**
(PDF)

## Acknowledgments

The Fiji Integrated Therapy study team for enrolment of participants, stool processing and KK microscopy in the field. *S. stercoralis*-positive stool was kindly provided by Professor Rob Baird with assistance from Dr. Joshua Osowicki. The vertebrate mitochondrial primer sequences, as well as *Ascaris lumbricoides* and *T. suis* eggs were provided to us by Professor Rebecca Traub.

## Author Contributions

**Conceptualization:** Kristy I. Azzopardi, Myra Hardy, Andrew C. Steer.

**Data curation:** Kristy I. Azzopardi, Myra Hardy, Ciara Baker, Rhian Bonnici.

**Formal analysis:** Kristy I. Azzopardi, Myra Hardy.

**Funding acquisition:** Myra Hardy, Andrew C. Steer.

**Investigation:** Kristy I. Azzopardi, Myra Hardy, Ciara Baker, Rhian Bonnici.

**Methodology:** Kristy I. Azzopardi, Myra Hardy, Stacey Llewellyn, James S. McCarthy, Rebecca J. Traub.

**Project administration:** Kristy I. Azzopardi, Myra Hardy.

**Resources:** Kristy I. Azzopardi, Myra Hardy, Rebecca J. Traub.

**Supervision:** Myra Hardy, Andrew C. Steer.

**Validation:** Kristy I. Azzopardi, Myra Hardy, Ciara Baker, Rhian Bonnici, Rebecca J. Traub.

**Visualization:** Kristy I. Azzopardi.

**Writing – original draft:** Kristy I. Azzopardi.

**Writing – review & editing:** Kristy I. Azzopardi, Myra Hardy, Ciara Baker, Rhian Bonnici, Stacey Llewellyn, James S. McCarthy, Rebecca J. Traub, Andrew C. Steer.

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
