## [Decision Letter · Decision Letter 0]

4 Aug 2021

PONE-D-21-19115

Detection of six soil-transmitted helminths in human stool by qPCR- a systematic workflow

PLOS ONE

Dear Dr. Azzopardi,

Thank you for submitting your manuscript to PLOS ONE. After careful consideration, we feel that it has merit but does not fully meet PLOS ONE’s publication criteria as it currently stands. Therefore, we invite you to submit a revised version of the manuscript that addresses the points raised during the review process.

The MS has improved substantially in this revised version. Look forward for the authors to address the comments from the Reviewers, especially the considerations made by the Reviewer 1.

We look forward to receiving your revised manuscript.

Kind regards,

Marcello Otake Sato, Ph.D., D.V.M.

Academic Editor

PLOS ONE

Journal Requirements:

2. Please clarify if the biological samples used in your study were:

(1) from an established biobank (if so please provide the name and a link)

(2) specifically collected for this study or not

(3) collected through a medically prescribed test

(4) completely de-identified before researchers accessed the samples

(5) if not de-identified, whether written informed consent was obtained.

3. We note that Figure 1 in your submission contain copyrighted images. All PLOS content is published under the Creative Commons Attribution License (CC BY 4.0), which means that the manuscript, images, and Supporting Information files will be freely available online, and any third party is permitted to access, download, copy, distribute, and use these materials in any way, even commercially, with proper attribution. For more information, see our copyright guidelines: http://journals.plos.org/plosone/s/licenses-and-copyright.

2. If you are unable to obtain permission from the original copyright holder to publish these figures under the CC BY 4.0 license or if the copyright holder’s requirements are incompatible with the CC BY 4.0 license, please either i) remove the figure or ii) supply a replacement figure that complies with the CC BY 4.0 license. Please check copyright information on all replacement figures and update the figure caption with source information. If applicable, please specify in the figure caption text when a figure is similar but not identical to the original image and is therefore for illustrative purposes only

Reviewers' comments:

Reviewer's Responses to Questions

**Comments to the Author**

1. Is the manuscript technically sound, and do the data support the conclusions?

Reviewer #1: Partly

Reviewer #2: Yes

2. Has the statistical analysis been performed appropriately and rigorously? 

Reviewer #1: Yes

Reviewer #2: N/A

3. Have the authors made all data underlying the findings in their manuscript fully available?

Reviewer #1: Yes

Reviewer #2: Yes

4. Is the manuscript presented in an intelligible fashion and written in standard English?

Reviewer #1: Yes

Reviewer #2: Yes

5. Review Comments to the Author

Reviewer #1: The manuscript is well presented and written; the authors have addressed the major and minor comments by other reviewers and the manuscript has gone significant improvement as suggested. The study is potentially important in providing method on sample collection and diagnostic of soil transmitted helminths.

Two major comments:

1. Although potassium dichromate is an appropriate stool preservative in tropical climates, I am not convinced that use of potassium dichromat is better than the use of other preservative such as alcohol/ethanol. Why is potassium dichromate for preservation was ideal for the trial location (Fijian islands)? Would that be also ideal to other endemic settings?

In the article van Papaikouvoe et al. Plos NTD 2018, that is cited by the author, is mentioned that after considering various factors such as preservative cost, inhibitor resistance, toxicity, availability, associated labor, sample shipping requirements, the use of 95% ethanol as a preservative is recommended in most situations. https://www.ncbi.nlm.nih.gov/pmc/articles/PMC5773002/

A study by Kaisar et al (Parasitology et al 2017) https://pubmed.ncbi.nlm.nih.gov/28290266/ and ten Hove et al (Trans Roy Soc 2008) https://pubmed.ncbi.nlm.nih.gov/18177680/ also showed that ethanol preserved stool can give good result and can be transported from the rural setting to the central laboratory.

And I agree with the authors that combination of DNA extraction and bead-beating can optimize the result, especially for T. trichiura eggs as shown in the study of Kaisar: The percentage of positive result in Trichuris PCR is increased when using ethanol preservation (45·0%), bead-beating (51·7%) and a combination (55·0%) and all three methods showed significantly higher DNA loads.

The study by Ayana et al showed that DNA concentration in samples preserved in potassium dichromate where lower compared to that found in sample preserved in ethanol and RNAlater. However, the gain in detecting cases was more pronounced for T. trichiura and potassium dichromate. https://journals.plos.org/plosntds/article?id=10.1371/journal.pntd.0007778

2. I wonder why there were no detection of Trichuris in the PCR? (S3 table) Would it be the negative effect of the preservation as mentioned by the author (e.g., multiple rinse process)? Was there explanation of the negative PCR results on positive KK for hookworm and Ascaris?

From 6 positive KK results, 3 have >160 EPG. It might be interesting to check in the ongoing study of the authors, whether use of other preservation such as ethanol, can prevent the negative findings. Is the prevalence of Trichuris low in the area (Fijian islands)? If the prevalence low, why use the stool samples from area to test the diagnostic tools and not looking for another region? Or test the prevalent helminths from the region.

The author mentioned that work is currently underway to test a larger sample size. I would prefer to wait for the result to have is published here.

Minor comment:

1. figure 4 needs better quality.

2. line 151: Samples were transported and stored at room temperature at The Murdoch Children’s Research Institute (MCRI) laboratories for testing within 2 years of collection.

I wonder whether the 40 samples were collected in the spread of 2 years, or the samples were hold in the field for 2 years? Or I misunderstood the statement?

Reviewer #2: Minor comment only:

Abstract: Line 33, bead beater was already mentioned in line 27, delete sentence

6. PLOS authors have the option to publish the peer review history of their article (what does this mean?). If published, this will include your full peer review and any attached files.

Reviewer #1: **Yes: **Aprilianto Eddy Wiria

Reviewer #2: No

---

## [Author Response · Author response to Decision Letter 0]

2 Sep 2021

Editor comments with author response

and 

Figure file names changed according to guidelines.

2. Please clarify if the biological samples used in your study were:

(1) from an established biobank (if so please provide the name and a link)

(2) specifically collected for this study or not

(3) collected through a medically prescribed test

(4) completely de-identified before researchers accessed the samples

(5) if not de-identified, whether written informed consent was obtained.

Biological samples used in this study were specifically collected for this study.

3. We note that Figure 1 in your submission contain copyrighted images. All PLOS content is published under the Creative Commons Attribution License (CC BY 4.0), which means that the manuscript, images, and Supporting Information files will be freely available online, and any third party is permitted to access, download, copy, distribute, and use these materials in any way, even commercially, with proper attribution. For more information, see our copyright guidelines: http://journals.plos.org/plosone/s/licenses-and-copyright.

We have acknowledged the source of copyrighted images depicted in Figure 1. (Line 240, added the sentence: “Images used under licence from Shutterstock.com”). All other icons are from Microsoft 365 and are free to use (no royalty or copyright). See link below for further information.

https://support.microsoft.com/en-us/topic/insert-icons-in-microsoft-office-e2459f17-3996-4795-996e-b9a13486fa79

Reviewer comments with author response

Reviewer's Responses to Questions

Comments to the Author

1. Is the manuscript technically sound, and do the data support the conclusions?

Reviewer #1: Partly

Reviewer #2: Yes

2. Has the statistical analysis been performed appropriately and rigorously?

Reviewer #1: Yes

Reviewer #2: N/A

3. Have the authors made all data underlying the findings in their manuscript fully available?

Reviewer #1: Yes

Reviewer #2: Yes

4. Is the manuscript presented in an intelligible fashion and written in standard English?

Reviewer #1: Yes

Reviewer #2: Yes

5. Review Comments to the Author

Reviewer #1:

The manuscript is well presented and written; the authors have addressed the major and minor comments by other reviewers and the manuscript has gone significant improvement as suggested. The study is potentially important in providing method on sample collection and diagnostic of soil transmitted helminths.

Two major comments:

1. Although potassium dichromate is an appropriate stool preservative in tropical climates, I am not convinced that use of potassium dichromat is better than the use of other preservative such as alcohol/ethanol. Why is potassium dichromate for preservation was ideal for the trial location (Fijian islands)? Would that be also ideal to other endemic settings?

In the article van Papaiakovou et al. Plos NTD 2018, that is cited by the author, is mentioned that after considering various factors such as preservative cost, inhibitor resistance, toxicity, availability, associated labor, sample shipping requirements, the use of 95% ethanol as a preservative is recommended in most situations. https://www.ncbi.nlm.nih.gov/pmc/articles/PMC5773002/

A study by Kaisar et al (Parasitology et al 2017) https://pubmed.ncbi.nlm.nih.gov/28290266/ and ten Hove et al (Trans Roy Soc 2008) https://pubmed.ncbi.nlm.nih.gov/18177680/ also showed that ethanol preserved stool can give good result and can be transported from the rural setting to the central laboratory.

And I agree with the authors that combination of DNA extraction and bead-beating can optimize the result, especially for T. trichiura eggs as shown in the study of Kaisar: The percentage of positive result in Trichuris PCR is increased when using ethanol preservation (45·0%), bead-beating (51·7%) and a combination (55·0%) and all three methods showed significantly higher DNA loads.

The study by Ayana et al showed that DNA concentration in samples preserved in potassium dichromate where lower compared to that found in sample preserved in ethanol and RNAlater. However, the gain in detecting cases was more pronounced for T. trichiura and potassium dichromate. https://journals.plos.org/plosntds/article?id=10.1371/journal.pntd.0007778

At the time that this study was conceived (2016), potassium dichromate was established as a reliable preservative of stools in this setting [1]. Cost was a major factor when choosing potassium dichromate over ethanol as stated in our manuscript (lines 106-110):

“Ethanol has been validated for use as a preservative in the absence of cold chain [2-5], but transport in large volumes by air becomes restricted and costly due to its classification as a dangerous good. Potassium dichromate is commonly used in STH field trials to provide long-term preservation of stool at ambient temperatures and does not have the same freight restrictions [1, 4, 6].”

Later on, publications by Papaiakovou et al in 2018 [4] and Ayana et al in 2019 [2] support the use of ethanol as a preservative of stool for detection of T. trichiura. For future studies, we would consider alternative preservatives as mentioned in our Discussion (lines 425-427):

“In future, we could trial different DNA and RNA preservatives that are commercially available. These reagents are available in kits, are nontoxic, and can be easily deployed to a remote setting with tropical climates, but they are likely to be more expensive.”

2a. I wonder why there were no detection of Trichuris in the PCR? (S3 table) Would it be the negative effect of the preservation as mentioned by the author (e.g., multiple rinse process)? 

Yes, this is a possibility that we mentioned in lines 428-430:

“The requirement for multiple rinses to remove the potassium dichromate may have also resulted in loss of free helminth DNA from degraded eggs, however most would be kept intact within eggs or larvae until DNA extraction was performed.”

For other possible reasons as to why we did not detect Trichuris by qPCR, see section 2c below.

2b. Was there explanation of the negative PCR results on positive KK for hookworm and Ascaris?

Yes, lines 408-413:

“Parasites detected by KK but not qPCR might be attributed to the over 4-fold increase in stool volume used for KK (41.7 mg) compared to qPCR (10 mg). We tried to limit this effect by maximising the volume of DNA tested (5 ul per reaction, in duplicate) in our multiplex qPCRs to achieve the highest sensitivity without compromising the reaction by introducing inhibitors.”

2c. From 6 Trichuris positive KK results, 3 have >160 EPG. It might be interesting to check in the ongoing study of the authors, whether use of other preservation such as ethanol, can prevent the negative findings. 

Yes, we discuss this in lines 418-427:

“A possible limitation in this study is the use of potassium dichromate as a preservative. It is important to note that in the Myanmar study, which reported higher detection of T. trichiura by qPCR than KK, stool was placed directly at -20°C and not preserved prior to molecular analysis [7]. It is possible that the potassium dichromate in our study could have prevented detection of T. trichiura by qPCR, as the effects of this preservative was not accounted for in our T. suis experiment. When compared to potassium dichromate, RNA later and 96% ethanol have been reported to increase DNA yield for A. lumbricoides, T. trichiura and N. americanus [2]. In future, we could trial different DNA and RNA preservatives that are commercially available. These reagents are available in kits, are nontoxic, and can be easily deployed to a remote setting with tropical climates, but they are likely to be more expensive.”

2d. Is the prevalence of Trichuris low in the area (Fijian islands)? If the prevalence low, why use the stool samples from area to test the diagnostic tools and not looking for another region? Or test the prevalent helminths from the region. The author mentioned that work is currently underway to test a larger sample size. I would prefer to wait for the result to have is published here.

Data documenting STH infections in Fiji is limited. A study in 2005 of Fijian school children showed the prevalence of Trichuris by microscopy to be 17% [8].

Minor comment:

1. figure 4 needs better quality.

We have increased the resolution of Figure 4.

2. line 151: Samples were transported and stored at room temperature at The Murdoch Children’s Research Institute (MCRI) laboratories for testing within 2 years of collection.

I wonder whether the 40 samples were collected in the spread of 2 years, or the samples were hold in the field for 2 years? Or I misunderstood the statement?

Sorry for the confusion. The sentence has been amended for clarification. Lines 151-154:

“Samples were collected over a period of 16 months (July 2017 to November 2018) and then transported to The Murdoch Children’s Research Institute (MCRI) laboratories where testing was performed between May 2018 to February 2019.”

Reviewer #2: 

Minor comment only:

Abstract: Line 33, bead beater was already mentioned in line 27, delete sentence

We have deleted the words “using the Mini-Beadbeater-24” (line 33).

6. PLOS authors have the option to publish the peer review history of their article (what does this mean?). If published, this will include your full peer review and any attached files.

Do you want your identity to be public for this peer review? For information about this choice, including consent withdrawal, please see our Privacy Policy.

Reviewer #1: Yes: Aprilianto Eddy Wiria

Reviewer #2: No

References

1. Llewellyn S, Inpankaew T, Nery SV, Gray DJ, Verweij JJ, Clements AC, et al. Application of a Multiplex Quantitative PCR to Assess Prevalence and Intensity Of Intestinal Parasite Infections in a Controlled Clinical Trial. PLoS Negl Trop Dis. 2016;10(1):e0004380. Epub 2016/01/29. doi: 10.1371/journal.pntd.0004380. PubMed PMID: 26820626; PubMed Central PMCID: PMCPMC4731196.

2. Ayana M, Cools P, Mekonnen Z, Biruksew A, Dana D, Rashwan N, et al. Comparison of four DNA extraction and three preservation protocols for the molecular detection and quantification of soil-transmitted helminths in stool. PLoS Negl Trop Dis. 2019;13(10):e0007778. Epub 2019/10/29. doi: 10.1371/journal.pntd.0007778. PubMed PMID: 31658264; PubMed Central PMCID: PMCPMC6837582.

3. Kaisar MMM, Brienen EAT, Djuardi Y, Sartono E, Yazdanbakhsh M, Verweij JJ, et al. Improved diagnosis of Trichuris trichiura by using a bead-beating procedure on ethanol preserved stool samples prior to DNA isolation and the performance of multiplex real-time PCR for intestinal parasites. Parasitology. 2017;144(7):965-74. Epub 2017/03/16. doi: 10.1017/S0031182017000129. PubMed PMID: 28290266; PubMed Central PMCID: PMCPMC5471844.

4. Papaiakovou M, Pilotte N, Baumer B, Grant J, Asbjornsdottir K, Schaer F, et al. A comparative analysis of preservation techniques for the optimal molecular detection of hookworm DNA in a human fecal specimen. PLoS Negl Trop Dis. 2018;12(1):e0006130. Epub 2018/01/19. doi: 10.1371/journal.pntd.0006130. PubMed PMID: 29346412; PubMed Central PMCID: PMCPMC5773002.

5. Song SJ, Amir A, Metcalf JL, Amato KR, Xu ZZ, Humphrey G, et al. Preservation Methods Differ in Fecal Microbiome Stability, Affecting Suitability for Field Studies. mSystems. 2016;1(3). Epub 2016/11/09. doi: 10.1128/mSystems.00021-16. PubMed PMID: 27822526; PubMed Central PMCID: PMCPMC5069758.

6. Stracke K, Clarke N, Awburn CV, Vaz Nery S, Khieu V, Traub RJ, et al. Development and validation of a multiplexed-tandem qPCR tool for diagnostics of human soil-transmitted helminth infections. PLoS Negl Trop Dis. 2019;13(6):e0007363. Epub 2019/06/18. doi: 10.1371/journal.pntd.0007363. PubMed PMID: 31206520.

7. Dunn JC, Papaiakovou M, Han KT, Chooneea D, Bettis AA, Wyine NY, et al. The increased sensitivity of qPCR in comparison to Kato-Katz is required for the accurate assessment of the prevalence of soil-transmitted helminth infection in settings that have received multiple rounds of mass drug administration. Parasit Vectors. 2020;13(1):324. Epub 2020/06/26. doi: 10.1186/s13071-020-04197-w. PubMed PMID: 32580759; PubMed Central PMCID: PMCPMC7315547.

8. Thomas M, Woodfield G, Moses C, Amos G. Soil-transmitted helminth infection, skin infection, anaemia, and growth retardation in schoolchildren of Taveuni Island, Fiji. N Z Med J. 2005;118(1216):U1492. Epub 2005/06/07. PubMed PMID: 15937527.

9. Kim SH, Stothard JR, Rinamalo M, Rainima-Qaniuci M, Talemaitoga N, Kama M, et al. A first nation-wide assessment of soil-transmitted helminthiasis in Fijian primary schools, and factors associated with the infection, using a lymphatic filariasis transmission assessment survey as surveillance platform. PLoS Negl Trop Dis. 2020;14(9):e0008511. Epub 2020/09/26. doi: 10.1371/journal.pntd.0008511. PubMed PMID: 32976499; PubMed Central PMCID: PMCPMC7518615.

---

## [Editor Report · Decision Letter 1]

17 Sep 2021

Detection of six soil-transmitted helminths in human stool by qPCR- a systematic workflow

PONE-D-21-19115R1

Dear Dr. Azzopardi,

We’re pleased to inform you that your manuscript has been judged scientifically suitable for publication and will be formally accepted for publication once it meets all outstanding technical requirements.

Kind regards,

Marcello Otake Sato, Ph.D., D.V.M.

Academic Editor

PLOS ONE

Additional Editor Comments (optional):

All the comments and suggestions by the Editor and Reviewers were satisfactorily completed by the authors. This MS is now ready for publication.
---

## [Editor Report · Acceptance letter]

23 Sep 2021

PONE-D-21-19115R1 

Detection of six soil-transmitted helminths in human stool by qPCR- a systematic workflow 

Dear Dr. Azzopardi:

I'm pleased to inform you that your manuscript has been deemed suitable for publication in PLOS ONE. Congratulations! Your manuscript is now with our production department. 

Kind regards, 

on behalf of

Dr. Marcello Otake Sato 

Academic Editor

PLOS ONE